# Spectroscopic and Microestructural Evidence for T-2 Toxin Adsorption Mechanism by Natural Bentonite Modified with Organic Cations

**DOI:** 10.3390/toxins15070470

**Published:** 2023-07-21

**Authors:** Fernando Abiram García-García, Eliseo Cristiani-Urbina, Liliana Morales-Barrera, Olga Nelly Rodríguez-Peña, Luis Barbo Hernández-Portilla, Cesar Mateo Flores-Ortíz

**Affiliations:** 1Laboratorio Nacional en Salud, Facultad de Estudios Superiores Iztacala, Universidad Nacional Autónoma de México, Av. de los Barrios No. 1, Tlalnepantla 54090, Mexico; abiramhiz@gmail.com (F.A.G.-G.); lbarbo@unam.mx (L.B.H.-P.); 2Departamento de Ingeniería Bioquímica, Escuela Nacional de Ciencias Biológicas, Instituto Politécnico Nacional, Av. Wilfrido Massieu s/n, Unidad Profesional Adolfo López Mateos, Ciudad de México 07738, Mexico; ecristianiu@yahoo.com.mx (E.C.-U.); lmoralesb@ipn.mx (L.M.-B.); 3Laboratorio de Fisiología Vegetal, Unidad de Biología, Tecnología y Prototipos (UBIPRO), Facultad de Estudios Superiores Iztacala, Universidad Nacional Autónoma de México, Av. de los Barrios No. 1, Tlalnepantla 54090, Mexico

**Keywords:** detoxifying agents, mycotoxins, natural aluminum silicate, physical detoxifying method, quaternary salts

## Abstract

Aluminosilicates are adsorbents able to bind mycotoxins, and their chemical modification increases their affinity to adsorb low-polarity mycotoxins. To further investigate if the inclusion of salts in bentonite modifies its adsorptive capacity, we studied T-2 toxin adsorption in natural bentonite (NB) and when modified with quaternary ammonium salts differing in polarity and chain length: myristyl trimethyl ammonium bromide (B14), cetyl trimethyl ammonium bromide (B16) and benzyl dimethyl stearyl ammonium chloride (B18). The results showed that quaternary salts made bentonite: displace monovalent (Na^+1^, K^+1^) and divalent (Mg^+2^, Ca^+2^) ions; reduce its porosity; change its compaction and structure, becoming more crystalline and ordered; and modify the charge balance of sheets. T-2 adsorption was higher in all modified materials compared to NB (*p* ≤ 0.0001), and B16 (42.96%) better adsorbed T-2 compared to B18 (35.80%; *p* = 0.0066). B14 (38.40%) showed no differences compared to B16 and B18 (*p* > 0.05). We described the T-2 adsorption mechanism in B16, in which hydrogen bond interactions, Van der Waals forces and the replacement of the salt by T-2 were found. Our results showed that interaction types due to the inclusion in B16 might be more important than the hydrocarbon chain length to improve the adsorptive capacity of bentonite.

## 1. Introduction

Mycotoxins are secondary metabolites mainly produced by fungi of the *Aspergillus*, *Penicillium* and *Fusarium* genera that differ in their chemical, biological and toxicological properties [1]. The *Fusarium* genus has been considered a cosmopolitan group of filamentous fungi with a high mycotoxin production mostly in the temperate regions of the northern hemisphere of America, Asia and Europe, causing severe infections in vegetables and fruits, and serious grain crop losses [2]. Aflatoxins, patulin, ochratoxins, zearalenone (ZEN) and trichothecenes have been widely recognized for their importance in public health and have been found as contaminants in grains, as well as processed and finished food, leading to important economic losses [3]. The T-2 toxin belongs to group A trichothecenes [4], it has been related to a great variety of animal and human toxicities, such as scabby grain toxicosis, alimentary toxic aleukia, Mseleni joint and Kashin-Beck diseases [5,6]. The epoxy group in its structure has been found to inhibit protein synthesis, damaging DNA and RNA synthesis, reducing lymphocyte proliferation and altering the dendritic cells maturation process [7]. Due to its toxicity, the Food and Agriculture Organization of the United Nations (FAO) and the European Union have established a tolerable daily intake of 60 ng kg^−1^ d^−1^ [8].

Given T-2 toxin toxicity, attempts have been made to treat food during storage, grinding or processing by biological, chemical, and physical processes [9]. While biological methods mainly operate through biochemical procedures by breaking the epoxide, biotransformation, genetic modification in fungi and grains and biological control of toxigenic strains, among others [10]; chemical methods have tried to reduce their contamination levels by using substances such as ammonia, formalin, calcium hydroxide, sodium bisulfite, ozone, chlorine and monomethylamine [11]; the physical ones focused on mechanical separation, such as flotation, removal of damaged grains, solution washing, irradiation and adsorption, besides others [12,13]. Adsorption involves the use of materials that attract and retain the mycotoxin on its surface, capable of not dissociating from the mycotoxin while passing through the host intestinal tract, in such a way that the adsorbent-mycotoxin complex can be eliminated through feces, minimizing the animal’s exposure to the toxin [14]. Therefore, the use of adsorbents offer a tool to ensure food safety and protect animals from mycotoxin levels, since they have particular structural and physicochemical traits such as particle size, specific area, pore size and charge, besides others, that influence their reactivity [15].

Aluminosilicates are nondigestible adsorbents, previously reported to be able to bind mycotoxins [16], and within types, differ in their bonding structure, ion exchange capacity, adsorption area, particle size and shape [17], formed by stacking of layers of tetra and octahedrons forming laminar structures. Substitutions between cations are produced in the tetra and octahedral layers that when they are of different valence, create a charge deficit, and to compensate for them, other cations are attracted and introduced between sheets [18]. Aluminosilicates are mainly composed of silicious and aluminum, on whose surface cations such as sodium, calcium and potassium are present. The ease of these cations to hydrate causes water molecules to group around them, providing a hydrophilic character to their surface, therefore they have a high affinity for polar compounds such as aflatoxins [19]. Bentonite, mainly composed of montmorillonite, has been recognized for its high ion exchange capacity [20] and its capacity to adsorb mycotoxins as aflatoxins, ochratoxins, sterigmatocystin, T-2 toxin and ZEN has been previously reported [21,22,23,24,25,26,27,28,29,30]. Moreover, chemical modification of aluminosilicates with ammonium salts has been found to increase their affinity and action spectrum [31], since they modify their structure by increasing the distance between laminar (L) and interlaminar (I) layers and pores, as well as their hydrophobicity increasing the adsorption of nonpolar molecules [31,32], allowing the aluminosilicate to better adsorb low polarity mycotoxins, such as ochratoxin A (OTA), T-2 toxin and ZEN [33,34,35,36].

To further investigate how the inclusion of ammonium salts in bentonite modifies adsorption, here, we aimed to study the T-2 toxin adsorption mechanisms of bentonite when tested at natural (NB) and when modified with three different quaternary ammonium salts differing in their polarity and chain length (14, 16 and 18 carbons): myristyl trimethyl ammonium bromide (MTAB; B14), cetyl trimethyl ammonium bromide (CTAB; B16) and benzyl dimethyl stearyl ammonium chloride (BDSAC; B18). To be able to compare original natural bentonite properties and those acquired with the quaternary salt’s modification, we first studied natural bentonite (NB) and modified tested materials (B14, B16 and B18) in their elemental composition, structure, porosity, charge balance, interlaminar space, electrostatic stability, and cation exchange capacity. Then we performed in vitro assays testing the T-2 toxin adsorption capability of all tested materials, to finally be able to propose an adsorption mechanism for the mycotoxin.

## 2. Results

### 2.1. Cation Exchange Capacity (CEC)

NB showed a total CEC of 506.39 meq/100 g, with a Na^+1^ dominance, indicating that it is a sodium bentonite. When saturating NB with NH_4_Cl solution, an exchange mainly occurred with monovalent cations, decreasing K^+1^ and Na^+1^ by 58.80 and 36.16%, respectively (Table 1).

### 2.2. Bentonite (NB) and Quaternary Salt-Modified Bentonite Materials Characterization

#### 2.2.1. Elemental Composition

The elements found in the greatest abundance in NB were Si, C and Al (Table 2). The differences in elemental composition between NB and all bentonite modified materials were found in all analyzed elements (*p* < 0.05). While concentration in C increased in all modified materials due to hydrocarbon chains inclusion of the quaternary ammonium salts, Al, Fe, Mg, Na and Si decreased. Ca^+2^ was detected in NB at a low concentration, but it was not present in all modified materials. Cl was only present in B18 compared to NB and the other two modified materials because it is found in the ion form in BDSAC. The presence of K^+1^ was not detected, so the monovalent ions displacement mainly occurs in Na^+1^.

#### 2.2.2. Charge Balance

NB showed negative charges in the octahedral sheets. The charge balance that occurs between the interlayer and the tetrahedral and octahedral sheets was 0.01, further indicating that all negative sites were compensated with positive charges of interlayer cations (Table 3). Ca^+2^ in modified materials became part of the interlayer cations that compensate for negative charges in the tetrahedral and octahedral sheets. For B14, MTAB inclusion caused positive charge loss in the tetrahedral sheet compared to NB, and a decrease in the octahedral sheet’s negative charges. The remaining charges were compensated by the cations still present in the interlayer, resulting in a charge balance of 0.01. For B16, CTAB increased negatively charged sites in the tetrahedral sheet when compared to NB, but they were lower compared to B14 (Table 3). Remaining charges were compensated by the cations in the interlayer, resulting in a charge balance of 0. B18 tetrahedral sheet charge was similar to NB. However, an increase in the negative charges in the octahedral sheet was observed, suggesting that the inclusion of BDSAC only took place in this sheet. Interlayer cations were not enough to offset the negative charges in the octahedral sheet, with a charge balance of −0.68 (Table 3).

#### 2.2.3. Porosity

Differences in apparent density were found (*F* = 47.53; *p* ≤ 0.0001) when the tested materials were compared (Figure 1). The results showed that the apparent density of B18 was significantly lower compared to NB (*p* = 0.0001), B14 and B16 (*p* ≤ 0.0001; Figure 1A). The real density was also found to be different among groups (*F* = 10.62; *p* = 0.0037); where all quaternary salt-modified bentonite materials had a lower real density, B14 (*p* = 0.0079); B16 (*p* = 0.0296) and B18 (*p* = 0.0039; Figure 1B) when compared to NB. Porosity was different among groups (*F* = 13.29; *p* = 0.0018); where all modified materials reduced their porosity compared to NB (*p* = 0.0015; 0.0132 and 0.0072 for B14, B16 and B18, respectively; Figure 1C).

#### 2.2.4. Morphological Changes

A disordered shape was found in NB with dispersed and separated irregular shaped particles (Figure 2). In contrast, micrographs of modified bentonite materials showed a change in its compaction and structure, being B18 with a more crystalline arrangement when compared to B14 and B16.

Based on the reduction in porosity in the bentonite modified with quaternary salt treatments compared to NB, it can be assumed that the phyllosilicate changes from a disorder state to a laminate arrangement in layers, which in turn, reduces porosity (Figure 3). Therefore, an interaction of the organic cations between the bentonite layers is assumed, providing greater symmetry and stability in the modified materials arrangement.

#### 2.2.5. Functional Group Analysis

The NB spectrum (Figure 4) showed the non-metallic mineral characteristic profile with intense signals at 917 cm^−1^ corresponding to Si-O bonds and vibrations at 3621 cm^−1^ corresponding to the structural OH, while 3413 cm^−1^ was assigned to bonds of water (O-H) in the interlayer. Furthermore, the signals of the organic cation in the modified materials showed the characteristic vibrations corresponding to N-H and long hydrocarbon chains. N-H bonds were found at 3400 cm^−1^, while the hydrocarbon chain signals were found at 2917 and 2848 cm^−1^, for C-H elongation of the CH_2_ in and out the plane, respectively. Moreover, the signal at 2872 cm^−1^ corresponds to C-H stretching of CH_3_. The vibrations at 1473 cm^−1^ refer to a lamellar ordering in the modified materials. The vibrations at 1465 and 1450 cm^−1^ were assigned to C-H bending of both CH_2_ and CH_3_ from long hydrocarbon chains. Finally, at 1430 cm^−1^ the signal corresponds to the ammonium ion and the overlapping of the bending vibration in the N-H plane at 1630 cm^−1^ were detected.

#### 2.2.6. ζ-Potential and Point of Zero Charge

Ζ-potential negative values were obtained for NB at all tested pH’s, indicating a charge compensation at the material surface by monovalent cations (Na^+1^; Figure 5) and not a point of zero charge was detected. In contrast, all modified materials showed a point of zero charge, allowing a great interaction with the T-2 toxin at these pH levels, where at the point of zero charge, the net electric charge on the material surface is zero, and the material surface is positively charged at pH’s below the zero charge point and negatively charged at pH’s above the zero charge point. B14 ζ potential values maintained negative charges in its surface from pH 4 to pH 9, since a charge compensation still occurring as in NB, and its point of zero charge was detected at pH 3.13. While the point of zero charge for B16 were found at pH’s 4.5 and 7. Moreover, for B18, as the pH increases, the ions interacting with the surface negative charges are released. In addition, ammonium ion substituents in B18 generated free negative charges that are not occupied either by monovalent or divalent ions such as Na^+1^, K^+1^, Ca^+2^ and Mg^+2^, or ammonium ions coming from the salt; its point of zero charge was detected at pH 4.25. Statistical differences in ζ-potential (mV) were found within materials (*F* (15, 40) =26.32; *p* ≤ 0.001). Tukey’s post-hoc tests showed differences between NB and all modified materials at pH’s 2, 4, 4.5, 6, 7 and 9 (significances range from *p* ≤ 0.05 to *p* ≤ 0.0001).

#### 2.2.7. T-2 Toxin and In Vitro Adsorption Assays

NB adsorbed the T-2 toxin by 15.09%. Significant differences were found when compared NB with the modified materials (*F* = 101.2; *p* ≤ 0.0001; Figure 6). All modified materials, B14, B16 and B18, adsorbed the T-2 toxin above 35%. Differences between B16 (42.96%) and B18 (35.80%) were found (*p* = 0.0066). No differences between B14 (38.40%) and the other two modified materials (B16 and B18) were found (*p* > 0.05).

#### 2.2.8. Mechanism of T-2 Toxin Adsorption

Although the inclusion of all quaternary salt types used here, increased T-2 toxin adsorption, in this section we chose to describe B16 adsorption mechanism since it exhibited the highest T2 toxin adsorption efficiency.

A homogeneous space in the interlaminar zone can be appreciated in NB (Figure 7A), due to CTAB inclusion in bentonite, two layers of tetrahedrons that encompass one of octahedrons laminar structure was observed (Figure 7B), and an increase in lamellar distances was found (*F* = 548.4; *p* ≤ 0.0001; Figure 8A). Sidak’s post hoc tests showed statistical differences in the bilayer space (*t* = 7.54; *p* ≤ 0.0001); intralamellar distance (distance between two interlaminar zones and a lamina; *t* = 14.70; *p* ≤ 0.0001) and interlamellar distance (distance between three interlaminar zones and two laminae; t = 22.04; *p* ≤ 0.0001).

When the T-2 toxin saturated the materials, all lamellar distances were reduced in both NB and B16, however, the interlamellar distance remains larger in B16 (0.79 nm) compared to NB (0.54 nm; t = 3.74; *p* = 0.0013; Figure 8B) and no differences in the interlayer space and in the intralamellar distance were detected. 

Although a homogeneous space in the interlaminar in NB was found (Figure 9A), a decrease in its interlamellar distance in NB when adsorbing the T-2 toxin may be because its inclusion could attract the upper or adjacent laminae (Figure 9A). However, in the case of B16, since the organic salt is included in the bentonite, a contact exists between the organic salt and the T2- toxin, where the arrangement within the interlayer is modified, making it disordered, as the chains accommodation imply different interaction types between molecules, where an exchange of the T-2 toxin takes the place of the organic salt. As T-2 is a smaller molecule compared with the organic salts, it occupies less space in B16, reducing all lamellar distances (Figure 9B).

The Figure 10 shows the infrared spectra of the T-2, NB, B-16 and B-16 + T-2. The signals observed at 2848 cm^−1^ and 2917 cm^−1^ were assigned to the CH_3_ and CH_2_, respectively, to the long chains of the CTAB, these signals are reduced in %T in the presence of T-2 toxin, suggesting the displacement of the organic cation by T-2 toxin. In addition, we can see the vibration at 1727 cm^−1^ which corresponds to ester carbonyls in the structure of T-2 toxin. The vibration at 1645 cm^−1^ belongs to the interaction with the free charges present in the Lewis acid sites with the ester carbonyls of T-2 toxin. The vibration at 1469 cm^−1^ refers to a C-H_2_ signal deformation related to the C-H_2_ hydrocarbon chains in the T-2 toxin, suggesting an interaction by electrostatic forces with CTAB.

## 3. Discussion

The binding efficacy of mineral adsorbents has been widely associated with both binders and the mycotoxins structures [13]. Since bentonites are considered as low-cost, eco-friendly and highly efficient in the adsorption of mycotoxins, they have been contemplated as promising mycotoxin adsorbents for animal feed [35,37,38]. Further, their modification could help to increase their adsorptive ability to non-polar mycotoxins such as T-2 toxin. Here we found that quaternary salts inclusion caused bentonite to: (i) displace monovalent (Na^+1^ and K^+1^) and divalent (Mg^+2^ and Ca^+2^) ions; (ii) reduce its porosity; (iii) change its compaction and structure making it more crystalline and ordered; (iv) modify the charge balance of sheets, and (v) all modified materials increased T-2 adsorption compared to NB. However, B16 was more effective in adsorbing T-2 compared to B-18, despite B18 has longer carbon chains. In this section, we will discuss our most relevant results.

The results obtained by ICP-MS when saturating NB with the NH_4_Cl solution showed there was an exchange mainly with monovalent cations, decreasing K^+1^ and Na^+1^, which agrees with those previously reported by Lagaly [39], who determined that the cations susceptible to being replaced by divalent ions or NH_4_^+^ is given in the series: Li^+1^ < Na^+1^ < H^+1^ < K^+1^ < NH_4_^+^ < Mg^+2^ < Ca^+2^. The presence and replacement of the divalent cations found in this study are similar to those reported by Li and Schulthess [40].

Since the presence of K^+1^ was not detected, the monovalent ion displacement mainly occurred in Na^+1^. The absence of K^+1^ in the interlaminar layer suggests that the divalent Ca^+2^ cation compensates the negative charges in the tetrahedral and octahedral layers, being displaced by the quaternary ammonium salts. The cation Ca^+2^ is usually present in the interlaminar zone when K^+1^ and Na^+1^ are found at low concentrations or are not present [41].

The results showed a loss of positive charges in the tetrahedral sheet and a decrease in negative charges in the octahedral sheet in B14. Long chain quaternary ammonium salts fixation has been found to mainly occur in the octahedral sheet, with the possibility to also randomly occur in the tetrahedral sheet [42], which mainly explain the charge decreases in both sheets. Despite the increase in negative charges in the tetrahedral sheet found here, the remaining charges were compensated by the cations still present in the interlayer reflected in the 0.01 charge balance obtained value. In the case of B16, we found that the salt inclusion increased negatively charged sites in the tetrahedral sheet compared to NB, but it was lower than B14 with a balance of 0. Similarly, He et al. [43], when modifying a clay rich in montmorillonite with this salt, found an irregular and random accommodation between the tetrahedral and octahedral sheets, which results in the negative charges decrease at the tetrahedral sheet. Related to B18, the results showed an increase in the negative charges at the octahedral sheet, which may be due to the presence of a benzene in one of its substituents which might be occupying a greater space, thus, displacing more than a hydrated cation in the octahedral sheet. Similarly, Bujdàk et al. [44] found that long chain quaternary ammonium salts mainly include in the octahedral sheet interacting with adjacent sheets.

Moreover, the exchangeable cations that compensate the charges present within the structure of the materials are in their hydrated form, so when the quaternary salts were included a decrease in the OH of the water of the surrounding exchangeable cations occurred, due to the monovalent cations’ removal and, in some cases, divalents. Further, the inclusion of the salts generated a change in the structural OH due to their interaction with the amino group from the quaternary ammonium salts. Moreover, the signal corresponding to the ammonium ion and the decrease in the OH in water surrounding the exchangeable cations indicated the inclusion of the quaternary salts in the bentonite structure, therefore changing the polarity from hydrophilic to hydrophobic, suggesting its anchorage within bentonite structure, which agree with previous studies [45,46,47,48,49,50].

B18 changed to have a more crystalline arrangement compared to NB, B14 and B16. It is possible that regarding the NH_4_^+^ substituent in each quaternary ammonium salt used, the arrangement in either tetrahedral or octahedral sheets will change. Similarly, Lazo et al., [51] modified an aluminosilicate with benzyl triethyl ammonium and tetramethyl ammonium, both with a NH_4_^+^ substituent similar to those used in this study, and found crystalline aggregates, depending on the NH_4_^+^ substituent further showing a more defined arrangement when the quaternary salt contained benzene. Additionally, an increase in the interlaminar space with a swollen appearance when including a quaternary salt to bentonite has been recognized [52].

It was found that all bentonite quaternary salt modified materials increased T-2 adsorption compared to NB. The inclusion of organic molecules in bentonite modifies their structure, conferring the property of adsorbing non-ionic organic compounds by hydrophobic interactions that depend on the acquired materials polarity, as well as on nitrogen substituents and chain length [53]. T-2 toxin, being a non-polar molecule, promotes Van der Waals interactions with the hydrocarbon chains in modified bentonite, and since long-chain quaternary salts confer an electrostatic charge that better attracts more neutrally molecules (non-polar), undoubtedly make T-2 toxin more susceptible to being adsorbed by modified bentonite [54]. Similar results were obtained by Nešić et al. [53], when a clinoptilolite adsorbent modified with esterified glucomannans was used. Our study clearly showed that bentonite modification with quaternary salts increases T-2 adsorption. Since the length of the hydrocarbon chains is positively related to salts polarity, the B16 salt polarity was the most capable to interact in a better way with the polar parts of the toxin. Moreover, since B14 and B16 present a similar molecular structure, and in contrast, B18 owns a benzyl group that provides the molecule with another polarity, allowing it to easily cross bentonite interlayers, the adsorption is less when compared to B16. Our results make evident the need to design studies testing modified bentonite adsorption capacity considering other trichothecenes found in the same *Fusarium sporotrichioides* biosynthesis pathway such as HT-2 toxin, diacetoxiscirpenol (DAS) and neosolaniol (NEO).

In this study an adsorption mechanism for B16 was proposed considering three different bonding types, where the hydrogen bonds would be present between the ester groups since the acetyl group in T-2 toxin would interact with bentonite Lewis acid site; creating electrostatic substitutions capable of removing and replacing the molecule with another with a greater force; Van der Walls interactions would generate a magnetic field among chains and therefore a dipole moment attraction where both attract and repel at the same time; and Lewis acid sites which shows hydrogen bonds formation. It has previously been reported that the hydrogen bonds between mycotoxins and Lewis acid sites of aluminosilicates are the main interaction for the adsorption, such as aflatoxin B1 carbonyls and the exchangeable cations in smectic [25]. The inclusion of an 18 C hydrocarbon chain quaternary salt has been found to be able to form a network between adjacent octahedral layers, attracting the T-2 toxin by electrostatic forces [26,55]. Further, zearalenone (ZEA) adsorption with a zeolite material modified with a quaternary salt was found to be due to Van der Waals forces [56]. In this study, due the 16-carbon low chain polarity and the T-2 toxin, interactions by Van der Walls were also more probable to occur. Moreover, the increase in interlaminar zone in B16 and its decrease when interacting with the T-2 toxin, further support that Van der Walls interaction play an important role in T-2 toxin adsorption. Finally, the decrease in the CTAB hydrocarbons signals in the infrared spectra (CH_2_ and CH_3_) suggest and exchange mechanism, in which the T-2 toxin replace the organic cation instead an adsorption in the low polarity network of the modified material. Future adsorption studies should include the trichothecenes belonging to the T-2 toxin group, such as diacetoxicirpenol, HT-2 toxin and neosolaniol (NEO).

## 4. Conclusions

Overall, T-2 toxin adsorption values reported in this study allow us to recognize the use of quaternary salt modified bentonite as an option to remove low polarity mycotoxins. Additionally, the results obtained in this study established that the main mechanism for adsorption of T-2 toxin on aluminosilicates occur through the polar interaction with the acid Lewis sites of the NB and through substitution of the organic cation in modified materials.

Additionally, we recognize that the kinetic and thermodynamic studies to better understand T-2 toxin adsorption processes, and the study of non-polar mycotoxins adsorption are recommended to subsequently propose the use of modified bentonite in feed and finished food for livestock as alternative management methods of contaminated raw materials.

## 5. Materials and Methods

### 5.1. Bentonite Preparation

Bentonite was provided by the Biogeochemical Laboratory, FESI, UNAM, México, and pulverized using the tribochemical process at the Green Chemistry Laboratory, FESC, UNAM, México. In brief, 20 g of bentonite were milled to small particles (5 mm diameter), and 4 g were placed in a 12.5 mL reactor equipped with three stainless steel spheres (1 cm diameter), finally, two of them were coupled in a high-speed planetary ball mill (Retsch^®^ PM100 2014, Haan, Germany) at 300 rpm and 28–30% power for 30 min. To obtain an approximate size of 75 µm, a 200 mesh was used to sift the sample (Montimex, Mexico City, Mexico). Natural bentonite pellets (500 mg) were prepared with a hydraulic press and stored until laboratory analyses were conducted.

### 5.2. Cation Exchange Capacity

One g of natural bentonite (NB) and 100 mL of 1 M NH_4_Cl were stirred at 300 rpm in an orbital shaker for 2 h and left to stand for 24 h. Samples were centrifuged at 10,000 rpm for 10 min and the pellet was recovered, lyophilized and digested following Kogel and Lewis [57]. In brief, a single solution containing cations with a final concentration of 1 mg/mL was prepared with NaCl, MgCl_2_, KCl and CaCl_2_. Data was analyzed by inductively coupled plasma mass spectrometry (ppm; ICP-MS; Agilent 7900; Model G8403A, Agilent, Tokyo, Japan) with MicroMist nebulizer, with a peristaltic pump and SPS 4 autosampler. Plasma ignition mode was used by aqueous solution. Scanning of five masses was performed (23Na, 24Mg, 39K and 43Ca). The nebulizer speed was 0.30 rps with a helium flow of 5.0 mL/min, OctP RF of 200 V, and discrimination power of 5.0 V.

### 5.3. Quaternary Salts Bentonite Modification

To chemically modify the natural bentonite with three cationic surfactants quaternary salts, we used: (1) myristyl trimethyl ammonium bromide MTAB (B14); (2) cetyl trimethyl ammonium bromide CTAB (B16); and (3) benzyl dimethyl stearyl ammonium chloride BDSAC (B18) by tribochemical process using the dry modification method [26,58,59,60,61,62]. In brief, 20 g of bentonite were homogenized in small rocks (5 mm diameter) and added to either: B14 (2.5 g NB + 0.798 g MTAB); B16 (2.5 g NB + 0.865 g CTAB) or B18 (2.5 g NB + 1.0073 g BDSAC) and pulverized using a stainless-steel reactor.

### 5.4. Bentonite (NB) and Quaternary Salt-Modified Materials Characterization

#### 5.4.1. Elemental Composition

Elemental analyses of tested materials for Al, C, Ca, Cl, Fe, K, Mg, Na and Si were carried out using X-ray fluorescence in a JEOL brand electron microscope (JSM-6380 LV; model 7582; JEOL, München, Germany) with X-ray accessories (InCA-sight EDS, Oxford Instruments, Abingdon, UK). Analyses were conducted in triplicate; data are presented as mean values (mean ± SD).

#### 5.4.2. Porosity

Porosity of tested materials was measured using the apparent and modified real density techniques with minor modifications proposed by Muñoz et. al., [63]. Analyses were conducted in triplicate; data are presented as mean values (mean ± SD).

#### 5.4.3. Scanning Electron Microscopy (SEM)

To assess for morphological changes due the quaternary ammonium salts inclusion, we analyzed samples with a scanning electron microscopy (SEM). The analysis was performed in a JEOL JSM-6380LV equipment, model 7582, equipped with an EDS InCA-sight Oxford Instruments X-ray accessory. Tested materials were prepared as follows: 500 mg of each tested material were weighed, and a pellet was made using a hydraulic press. Samples were covered with gold to improve resolution. Samples were stored until analysis.

#### 5.4.4. High Resolution Transmission Electron Microscopy (HR-TEM)

The analysis was performed at the Transmission Electron Microscopy Laboratory of the Center for Nanosciences and Microtechnology at the National Polytechnic Institute, using a JEOL JEM-21000 (Tokyo, Japan) microscope with LaB6 filament and acceleration voltages of 80 to 200 kV. In brief, an aliquot of each tested material was placed in a 2 mL vial with isopropyl alcohol. The control group, only containing tested materials were sonicated for 10 min, while the tested samples containing the mycotoxin were sonicated for 13 min. Then, a drop was taken with a Pasteur pipette and placed on a slide-holder and left to dry until analysis.

#### 5.4.5. Fourier Transform-Infrared Spectroscopy Analysis (FT-IR)

To determine the change in the structure of NB due to inclusion of quaternary ammonium salts and to be able to infer about action mechanisms, we used the Fourier Transform-Infrared Spectroscopy analysis (FT-IR) with a reflecting technique. Briefly, the ATR spectra were obtained using a Perking Elmer (FT-IR; Waltham, MA, USA) model Frontier FT-IR/NIR with a MIR TGS detector equipped with a diamond crystal and a MIR source, with a resolution of 4 cm^−1^ and 32 scans for each sample.

#### 5.4.6. Zeta Potential (ζ) and Point of Zero Charge

To assess the electrostatic stability and charges behavior in the solid-liquid interface of tested materials we analyzed the zero point of charge (pH^pzc^). Briefly, solutions containing water and 100 mg of the tested material were prepared at different pH conditions (2, 4, 4.5, 6, 7 and 9). Samples were vortexed for 1 min, and centrifuged at 5000 rpm. To measure ζ we used a Zetasizer equipment (Nano ZS; Malvern Zetasizer ZEN3600, Worcestershire, UK). While ζ is the potential difference between the dispersion medium and the stationary layer of the fluid, the point of zero charge (PZC) is the pH at which the overall charge of the particles is zero [64]. Analyses were conducted in triplicate. Data are presented as mean values (mean ± SD).

#### 5.4.7. T-2 Toxin In Vitro Adsorption Assays

The assay solution was prepared containing 30 ppm of theT-2 toxin in a 150 mM phosphate buffer at pH 7. In a final volume of 1 mL, T-2 toxin solution, 1 mg of NB or the modified materials were mixed. Adsorption assays were kept under mechanical horizontal agitation at 150 rpm, incubated at 41 °C for 60 min and centrifuged at 14,000 rpm for 3 min. Supernatant was used to quantify residual T-2. Adsorption assays were conducted in triplicate. Data are presented as mean values (mean ± SD).

#### 5.4.8. Residual T-2 Quantification

Residual T-2 toxin quantification was assessed by HPLC-ESI-TOF-MS (High Pressure Liquid Chromatography-Electrospray-Time of Flight-Mass Spectrometry) in reverse phase. In short, we used a positive ion mode HPLC system, consisting on a vacuum degasser, autosampler and binary pump (Infinity 1260, Agilent Technologies, Santa Clara, CA, USA); equipped with an Eclipse Plus column (C-18 RRHD, 1.8 µm, 2.1 × 100 mm; Agilent, Santa Clara, CA, USA). The column temperature was maintained at 25 °C. Mobile phase consisted of 10 mM of ammonium acetate (A) and HPLC-grade methanol (B); the gradient began with 85% A and 15% B, then changed to 100% B at minute 40 and was maintained for 10 min. Flow rate was 0.200 mL/min. The column equilibration time was 10 min. The injection volume was 5 µL. HPLC was coupled to TOF/MS (Time of Flight-Mass Spectrometry; Agilent 66230B; Yishun, Singapore) with electrospray interface. Operating conditions were as follows: gas temperature at 250 °C, gas flow at 6 l/min and nebulizer pressure at 60 psig, shredder at 200 V, skimmer at 65 V and OCT RF Vpp at 750 V. Data was acquired using Mass Hunter Workstation LC/MS Data acquisition for 6200 series software version B.06.01, Build 6.0.633.10 (2012, Agilent Technologies, Santa Clara, CA, USA).

### 5.5. Data Analyses

To assess for differences in elemental composition between the natural bentonite (NB) and the three modified materials, porosity and T-2 toxin adsorption, we performed one-way multiple analysis of variance (ANOVA) followed by Tukey’s (elemental composition, real density, porosity and T-2 toxin adsorption data), Sidak’s (lamellar distances) or Dunnett’s (apparent density data) post-hoc tests [65]. A two-way RM ANOVA followed by a Tukey’s test was performed to analyze Zeta potential (ζ) and to determine the point zero of charge. A *p* < 0.05 was considered significant. Analyses and figures were performed and created using Prism 8^®^ Version 8.4.0 for Mac OS (GraphPad Software, Inc., La Jolla, CA 92037, USA).

## Figures and Tables

**Figure 1 toxins-15-00470-f001:**
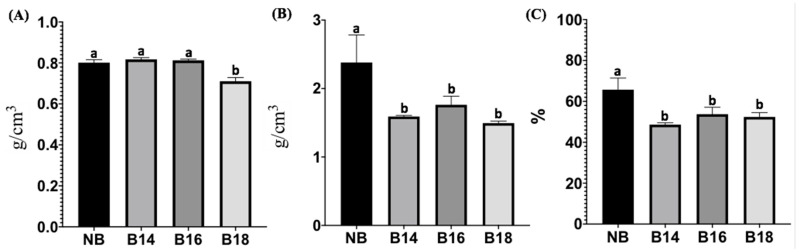
Density and porosity for natural bentonite modified materials. (**A**) Apparent density; (**B**) real density; (**C**) porosity. (NB) natural bentonite; (B14) bentonite + myristyl trimethyl ammonium bromide; (B16) bentonite + cetyl trimethyl ammonium bromide; (B18) bentonite + benzyl dimethyl stearyl ammonium chloride. Data were analyzed using one-way analysis of variance (ANOVA) followed by post-hoc tests. Data not sharing at least one letter are different (*p* < 0.05).

**Figure 2 toxins-15-00470-f002:**
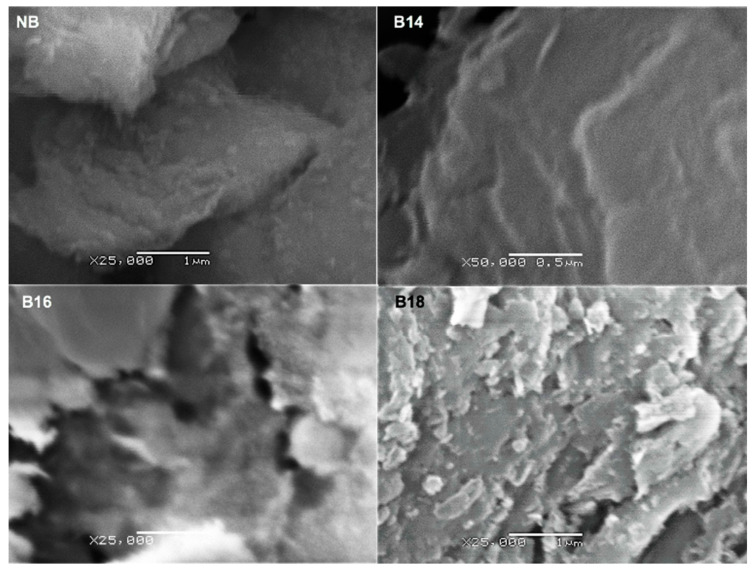
Scanning electron micrographs of Bentonite tested materials (SEM). (NB) natural bentonite; (B14) bentonite + myristyl trimethyl ammonium bromide; (B16) bentonite + cetyl trimethyl ammonium bromide; (B18) bentonite + benzyl dimethyl stearyl ammonium chloride.

**Figure 3 toxins-15-00470-f003:**
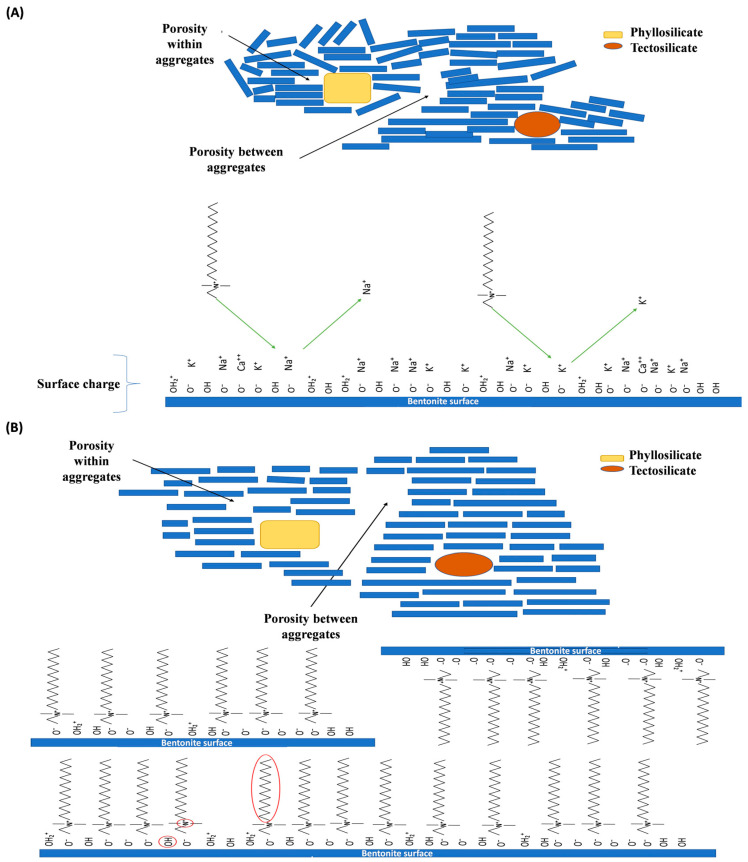
Graphic model describing the lamellar ordering transition found in modified materials compared to natural bentonite. (**A**) natural bentonite (NB) characteristic lamellar microstructure and displacement of monovalent ions on its surface by quaternary salts; (**B**) bentonite lamellar microstructure, which acquires greater ordering after including the quaternary ammonium salts.

**Figure 4 toxins-15-00470-f004:**
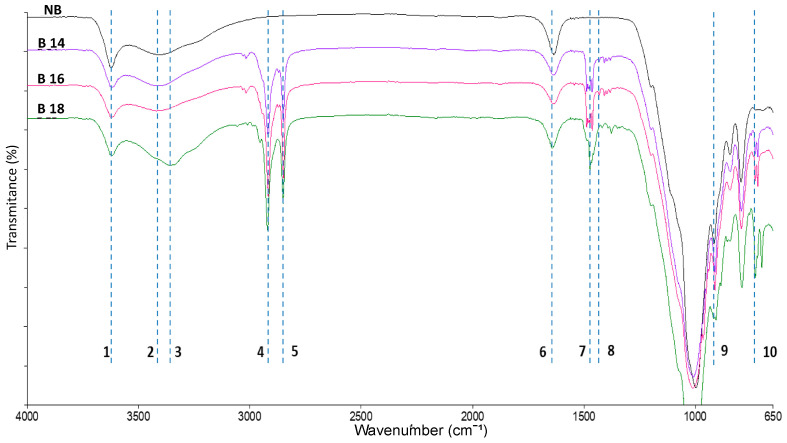
Mid-range infrared spectrum of natural bentonite and modified materials. (NB) natural bentonite; (B14) bentonite + myristyl trimethyl ammonium bromide; (B16) bentonite + cetyl trimethyl ammonium bromide; (B18) bentonite + benzyl dimethyl stearyl ammonium chloride: (1) 3621 cm^−1^; (2) 3413 cm^−1^; (3) 3349 cm^−1^; (4) 2917 cm^−1^; (5) 2848 cm^−1^; (6) 1631 cm^−1^; (7) 1473 cm^−1^; (8) 1431 cm^−1^; (9) 917 cm^−1^; (10) 720 cm^−1^.

**Figure 5 toxins-15-00470-f005:**
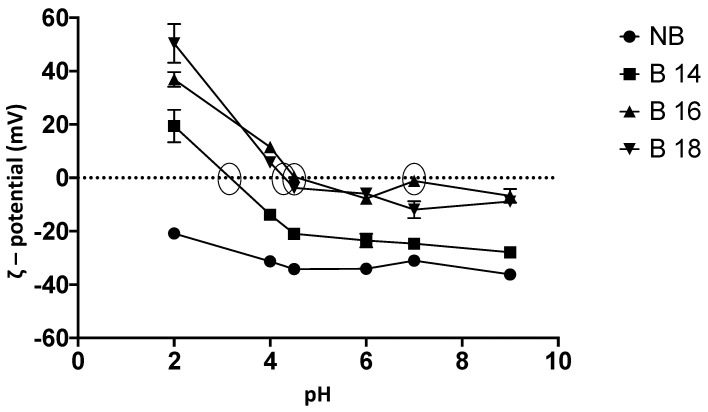
Zeta potential (ζ) and point of zero charge of tested materials. Circles in the dotted line (0 mV) show the point of zero charge, that correspond to B14 at pH 3.13, B16 at 4.5 and 7, and B18 at 4.25, indicating their most hydrophobic points that allow materials to better adsorb low polarity compounds such as the T-2 toxin. (NB) natural bentonite; (B14) bentonite + myristyl trimethyl ammonium bromide; (B16) bentonite + cetyl trimethyl ammonium bromide; (B18) bentonite + benzyl dimethyl stearyl ammonium chloride.

**Figure 6 toxins-15-00470-f006:**
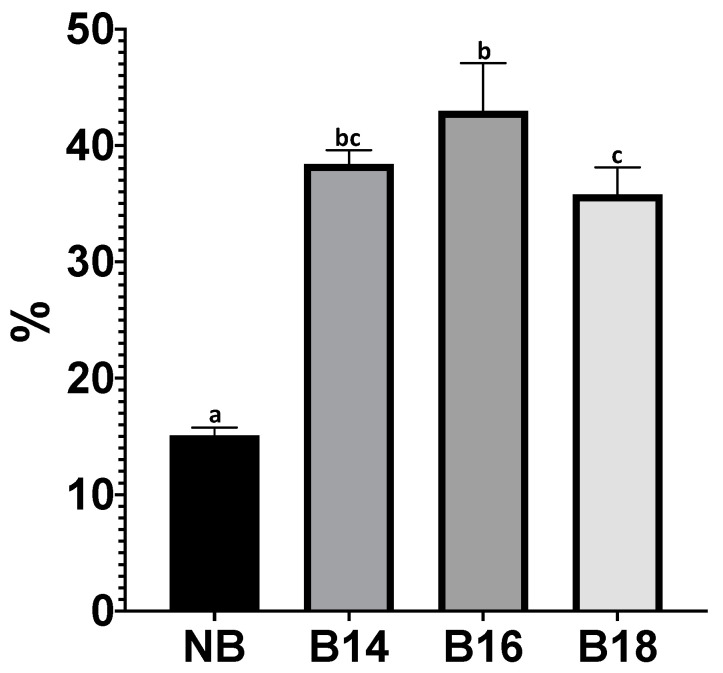
Adsorption (%) of T-2 toxin by tested materials. (NB) natural bentonite; (B14) bentonite + myristyl trimethyl ammonium bromide; (B16) bentonite + cetyl trimethyl ammonium bromide; (B18) bentonite + benzyl dimethyl stearyl ammonium chloride. Data not sharing at least one letter are different (*p* < 0.05).

**Figure 7 toxins-15-00470-f007:**
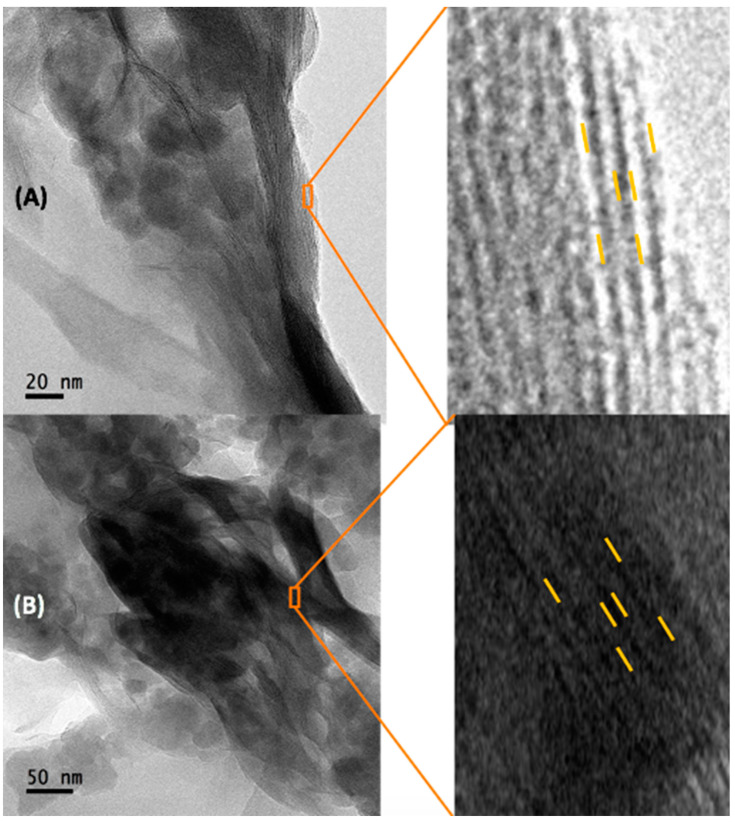
Micrographs obtained by Transmission Electron Microscopy (TEM) comparing interlaminar space of NB and B16. (**A**) natural bentonite close-up of the interlaminar zone; (**B**) B16 close-up of the interlaminar zone. (NB) natural bentonite; (B16) bentonite + cetyl trimethyl ammonium bromide.

**Figure 8 toxins-15-00470-f008:**
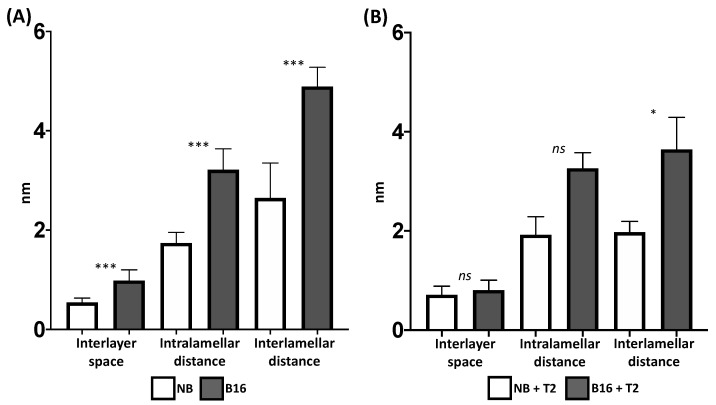
Natural bentonite and B16 lamellar distances comparison (**A**) without T-2 toxin and (**B**) saturated with T-2 toxin. (NB) natural bentonite; (B16) bentonite + cetyl trimethyl ammonium bromide *ns* = non-significant; * = *p* < 0.05; *** = *p* < 0.0001.

**Figure 9 toxins-15-00470-f009:**
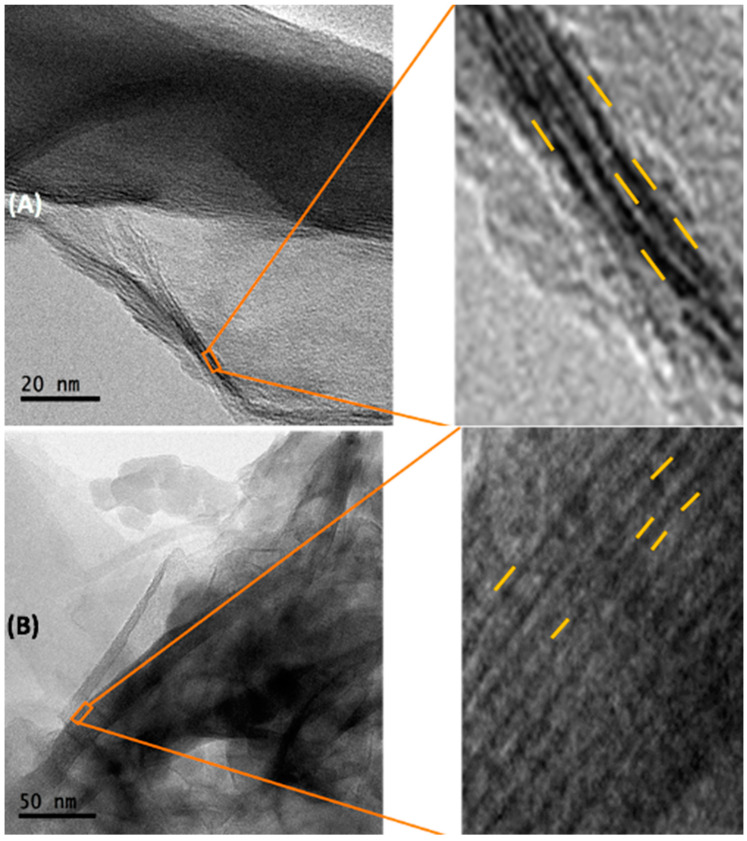
Micrographs obtained by Transmission Electron Microscopy (TEM) comparing interlaminar space of NB and B16 saturated with T-2 toxin (25 ppm). (**A**) Natural bentonite close-up of the interlaminar zone; (**B**) B16 close-up of the interlaminar zone. (NB) natural bentonite; (B16) bentonite + cetyl trimethyl ammonium bromide.

**Figure 10 toxins-15-00470-f010:**
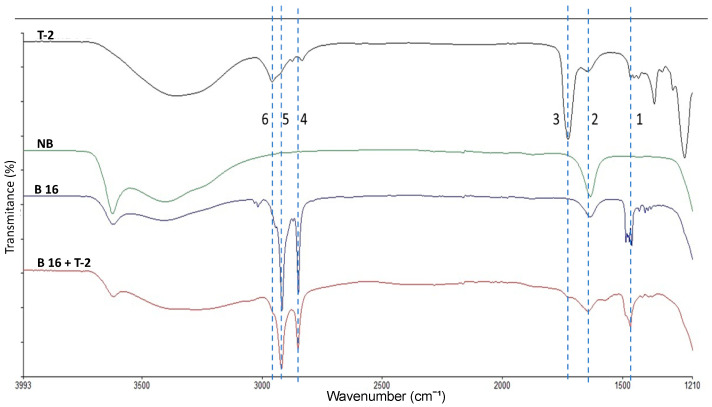
Mid-range infrared spectrum of T-2 toxin, natural bentonite, the cetyl trimethyl ammonium bromide bentonite modified material, with and without the T-2 toxin. (1) 1469 cm^−1^; (2) 1645 cm^−1^; (3) 1727 cm^−1^; (4) 2848 cm^−1^; (5) 2917 cm^−1^; (6) 2958 cm^−1^. (T-2) T-2 toxin; (NB) natural bentonite; (B16) bentonite + cetyl trimethyl ammonium bromide; (B16 + T-2) bentonite + cetyl trimethyl ammonium bromide + T-2 toxin.

**Table 1 toxins-15-00470-t001:** Natural bentonite (NB) cation exchange capacity.

meq/100 g					
Element (MW)	K (39)	Na (23)	Ca (43)	Mg (24)	Total
NB	3.52 ± 1.01	105.32 ± 8.83	54.51 ± 1.89	343.04 ± 1.83	506.39 ± 10.76
NB + NH_4_Cl	1.45 ± 2.36	67.21 ± 30.39	43.88 ± 9.21	304.89 ± 50.51	417.43 ± 99.38

**Table 2 toxins-15-00470-t002:** Elemental composition of tested materials.

Chemical Element	NB	B14	B16	B18	ANOVA (*F* Value; *p*)
Al	5.08 ± 0.36 ^a^	4.94 ± 0.29 ^a^	4.81 ± 0.44 ^a^	2.52 ± 0.38 ^b^	31.29; <0.0001
C	12.28 ± 1.39 ^a^	17.22 ± 0.66 ^b^	16.89 ± 0.39 ^b^	19.57 ± 0.08 ^b^	44.06; <0.0001
Ca	0.33 ± 0.03 ^a^	0 ^b^	0 ^b^	0 ^b^	275.7; <0.0001
Cl	0 ^a^	0 ^a^	0 ^a^	1.62 ± 0.06 ^b^	2019; <0.0001
Fe	0.84 ± 0.46 ^a^	0.59 ± 0.11 ^a^	0 ^b^	0 ^b^	9.73; 0.004
K	0	0	0	0	*ns*
Mg	0.94 ± 0.07 ^a^	0.66 ± 0.04 ^a^	0.65 ± 0.12 ^a^	0.31 ± 0.23 ^b^	10.07; 0.004
Na	0.90 ± 0.04 ^a^	0.51 ± 0.09 ^b^	0.51 ± 0.15 ^b^	0.39 ± 0.04 ^b^	17.32; 0.0007
Si	19.10 ± 1.83 ^a^	11.69 ± 0.71 ^b^	12.72 ± 0.86 ^b^	9.63 ± 0.42 ^b^	41.71; <0.0001

(NB) natural bentonite; (B14) bentonite + myristyl trimethyl ammonium bromide; (B16) bentonite + cetyl trimethyl ammonium bromide; (B18) bentonite + benzyl dimethyl stearyl ammonium chloride. Different superscripts showed statistical significances between groups. Analyses were conducted in triplicate; data are presented as mean values (mean ± SD). Data were analyzed using one-way analysis of variance (ANOVA) followed by Tukey’s post-hoc test. Data not sharing a letter are different (*p* < 0.05). *ns* = non-significant.

**Table 3 toxins-15-00470-t003:** Theoretical charge balance of natural bentonite and modified materials with quaternary salts.

Tested Material		NB	B14	B16	B18
Sheet charge	Octahedral	−0.93	−0.08	−0.26	−1.58
	Tetrahedral	0.56	−0.24	−0.05	0.56
	Interlayer	0.38	0.32	0.31	0.34
Layer charge		−0.37	−0.32	−0.31	−1.02
Charge Balance		0.01	0.01	0	−0.68

(NB) natural bentonite; (B14) bentonite + myristyl trimethyl ammonium bromide; (B16) bentonite + cetyl trimethyl ammonium bromide; (B18) bentonite + benzyl dimethyl stearyl ammonium chloride.

## Data Availability

The data presented in this study are available in this article.

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
