# Peer review of "Spectroscopic and Microestructural Evidence for T-2 Toxin Adsorption Mechanism by Natural Bentonite Modified with Organic Cations"

_toxins, 2023, doi:10.3390/toxins15070470_

Round 1
Reviewer 1 Report
In this research article entitled " Mechanisms of adsorption of T-2 toxin by natural bentonite and when modified with organic cations" the authors explored to study the T-2 toxin adsorption mechanisms of bentonite when tested at natural (NB) and when modified with three different quaternary ammonium salts (MTAB; B14, CTAB;B16 and BDSAC; B18). Quantitatively, there were performed enough experiments and results and discussion were presented and analysed very well. Tables and figures are mainly clear and organized. In my opinion, this is a very well prepared and written article. However, I mention below some points that should be considered before processing further.
· - write p value in italics as in the entire article (I recommend checking all document).
· - Figure 2 – does it seem to me or is this picture blurry (B18)? If you have a better one, please replace it.
· - chapter 2.2.6 there is a gap missing.
· - write in vitro in italics throughout the article.
· - there are big gaps on tent 3 and 13, please fix it.
In conclusion, the article is well written, a few adjustments need to be made in it, and after minor corrections it could be published in Toxins. I think you could add to the title, which is written very briefly for my taste. Highlight the quality of your study, add some limits and, of course, future perspectives.
Reviewer 2 Report
The manuscript is in good shape, and the data presented is of high scientific Interest. Only minor revisions to fix abbreviations are needed throughout the text, such as: put definitions before acronyms (e.g., L.69 - use "ZEN" instead of "ZEA"); L.74 "ochratoxin".
Reviewer 3 Report
The manuscript describes the use of bentonite clay material to remove T-2 toxin. The manuscript is well written and is acceptable in its present form.
Please add x and y - axis title to figure 4 and 10.
